# Position: Graph Learning May Have Been Misled By Over-smoothing And Over-squashing

## Abstract

The prevalent focus in graph learning research on theoretical challenges like over-smoothing and over-squashing may be misguiding, as their practical relevance in real-world scenarios is questionable. While Graph Neural Networks (GNNs) have achieved significant success across various applications, theoretical work has extensively discussed issues such as over-smoothing and over-squashing for the past eight years. This paper argues that the continued emphasis on these problems might be misplaced. For node-level tasks, we suggest that performance decreases often stem from uninformative receptive fields rather than over-smoothing, as optimal model depths remain small even with mitigation techniques. For graph-level tasks, over-smoothing can even be beneficial if the smoothed state is label-informative. Similarly, we challenge the notion that over-squashing, i.e., the compression of exponentially growing information into fixed-size node embeddings, is always detrimental in practical applications. We argue that the distribution of relevant information over the graph frequently factorises and is often localised within a small $k$-hop neighbourhood, questioning the necessity of jointly observing entire receptive fields or engaging in an extensive search for long-range interactions. Our empirical findings demonstrate that while methods exist to mitigate over-smoothing and over-squashing, they often do not yield significant performance gains with increased model depth on standard benchmarks, and can lead to substantial computational costs. This position paper advocates for a paradigm shift in theoretical research, urging a diligent analysis of future learning tasks and datasets to better understand the localisation and factorisation of label-relevant information. This will ensure that theoretical advancements align with the real needs and challenges of real-world graph learning problems.

## 1 Introduction

A graph is a versatile data structure that is well-suited for describing complex systems, offering a mathematical foundation to analyse complex real-world systems using Graph Neural Networks (GNN). Gilmer et al. (2017) introduced the Message Passing Neural Network (MPNN) framework in 2017, which serves as the foundation for most modern GNNs, with powerful variants like graph convolutional networks (GCNs) (Kipf and Welling, 2017) and graph attention networks (GATs) (Veličković et al., 2018; Brody et al., 2022) achieving state-of-the-art results on many graph learning tasks. These advances have spurred a wide range of applications. For example, GNNs are now routinely applied in chemistry and drug discovery for molecular property prediction (Li et al., 2021), in large-scale recommendation engines in social networks (Ying et al., 2018; Zhao et al., 2025; Liu et al., 2024), and in spatial–temporal forecasting for traffic prediction like for Google Maps (Yu et al., 2017; Derrow-Pinion et al., 2021). Theoretical work on GNNs has explored several challenges that limit their capabilities for graph learning tasks. Most prominently, the problems of over-smoothing,

over-squashing, robustness and expressivity are discussed in many publications over the past eight years (Rusch et al., 2023a; Akansha, 2025; Zhang et al., 2025; Morris et al., 2019; Zhang et al., 2024; Günnemann, 2022). Over-smoothing occurs when node representations become indistinguishable after many layers, typically measured through concepts like Dirichlet Energy. Over-squashing refers to the issue where node embeddings of constant size struggle to represent information of exponentially growing receptive fields, often conceptualised through Jacobian matrices. Robustness examines a GNNs sensitivity to perturbations in input features or graph structure, with research focusing on adversarial attacks and defences. Finally, expressivity investigates the theoretical power of GNNs to distinguish between different graph structures and learn complex functions, often explored by comparing them to the Weisfeiler-Lehman test. While addressing all four challenges remains an active focus of current GNN research, this paper critically examines over-smoothing and over-squashing.

In a recent position paper Bechler-Speicher et al. (2025) posit that Graph Representation Learning may lose relevance due to poor benchmarks. They criticise (1) the relevance and real-world impact of learning tasks on current benchmark datasets, (2) the suitability of certain benchmark datasets for the application of GNNs, and (3) the overall benchmarking culture. Consequently, they propose a paradigm shift towards driving impactful advances in graph learning research and call for new transformative real-world applications. We anticipate that this position paper will lead to an extensive search for such applications, potentially followed by introductions of new datasets and new learning tasks

**We believe diligent analysis of these learning tasks with statistics that measure the underlying distributions of relevant information is required to guide further directions of theoretical research. This position paper questions the practical relevance of over-smoothing and over-squashing in currently established learning tasks.**

**Preliminaries.** Let $\mathcal{G} = (\mathcal{V}, \mathcal{E})$ be an undirected graph with node set $\mathcal{V}$ and edge set $\mathcal{E}$. The graph $\mathcal{G}$ can be represented via an adjacency matrix $\boldsymbol{A} \in \mathbb{R}^{|\mathcal{V}| \times |\mathcal{V}|}$ with $\boldsymbol{A}_{v,u} \neq 0$ if and only if $(v, u) \in \mathcal{E}$. Each node $v \in \mathcal{V}$ has a neighbourhood $\mathcal{N}(v) = \{u : (v, u) \in \mathcal{E}\}$, a $k$-hop neighbourhood $\mathcal{N}^k(v) = \{u \in \mathcal{V} \mid d_G(v, u) \leq k\}$, where $d_G(v, u)$ denotes the shortest path distance between $v$ and $u$ in the graph $\mathcal{G}$, and features $\boldsymbol{h}_v^{(0)} \in \mathbb{R}^{d_0}$, which are placed on the rows of the matrix $\boldsymbol{H}^{(0)} \in \mathbb{R}^{|\mathcal{V}| \times d_0}$. Graph Neural Networks (GNNs) are neural networks designed to process graph-structured data. Most GNNs fall into the class of MPNNs, in which layers of message passing functions $M(\cdot)$ and update functions $U(\cdot)$ are iteratively composed. In layer $\ell$, a new hidden embedding is computed as

$$\boldsymbol{h}_v^{(\ell+1)} = U\left(\boldsymbol{h}_v^{(\ell)}, M\left(\boldsymbol{h}_v^{(\ell)}, \{\boldsymbol{h}_u^{(\ell)} : u \in \mathcal{N}(v)\}\right)\right).$$

To exemplify this abstract function class, we consider the model equation of the GCN (Kipf and Welling, 2017) below

$$\boldsymbol{H}^{(\ell)} = \sigma\left(\tilde{\boldsymbol{A}} \boldsymbol{H}^{(\ell)} \boldsymbol{W}^{(\ell)}\right),$$

where the update step is defined by the non-linearity $\sigma$ and learnable weight matrix $\boldsymbol{W}^{(\ell)}$, and the message passing step is implemented via the message passing operator $\tilde{\boldsymbol{A}} = (\boldsymbol{D} + \boldsymbol{I})^{-1/2}(\boldsymbol{A} + \boldsymbol{I})(\boldsymbol{D} + \boldsymbol{I})^{-1/2}$. Note that we omit trainable bias vectors here and will continue to do so throughout our discussion for brevity. For a MPNN, the receptive field of a node $v$ refers to the set of nodes whose initial features $\boldsymbol{h}^{(0)}$ influence the final embedding $\boldsymbol{h}_v^{(\ell)}$, i.e., all nodes with $d_G(v, u) \leq \ell$.

## 2 Over-smoothing

We believe that the continued focus of theoretical research on over-smoothing within the graph learning community constrains the practical relevance of recent contributions. In this section, we substantiate our position by first reviewing the extensive body of literature on over-smoothing, which in our view indicates that the phenomenon is both well-understood and effectively mitigated. Based on the literature, we call into question the importance of working with arbitrarily large receptive fields in current GNN research. We back our position with experimental results demonstrating that even GNNs designed to avoid over-smoothing do not yield significant performance gains on standard benchmarks.

## 2.1 Related Work

Over-smoothing refers to the phenomenon wherein node representations converge to a single representation with an increasing number of message passing steps and become therefore indistinguishable. In many publications, this is assumed to be the main reason for a relatively small optimal number of message passing layers compared to other neural architectures. Convolutional Neural Networks (CNNs), for example, gain performance by increasing the number of convolutions, resulting in CNN state-of-the-art architectures with hundreds of layers.

**Theory and Quantification of Over-smoothing.** Kipf and Welling (2017) first discovered a performance peak of GNNs, here the Graph Convolution Network (GCN), at around 7–8 message-passing layers. Li et al. (2018) further investigated this phenomenon and demonstrated empirically for a small graph dataset (Zachary's karate club network (Zachary, 1977)) that repeated graph convolutions first improve the class separation in the latent space before collapsing the representations. Based on the simplification of graph convolutions as a special form of Laplacian smoothing, they show that infinitely repeated applications of the message passing operator lead node embeddings to converge to a constant vector.

Chen et al. (2019) approached the problem of over-smoothing by measuring (over-)smoothness. They propose to use the Mean Average Distance (MAD) as a quantitative metric for the smoothness of the graph. The MAD applies the cosine distance to the representations of each node pair after the final layer $L$ of the GNN. Based on this distance, they define the MADGap, the difference between remote node pairs $\text{MAD}^{rmt}$ and neighbours $\text{MAD}^e$, as an over-smoothness metric. Their experiments suggest a connection between over-smoothness, performance, and the information-to-noise ratio, quantified by the proportion of same-class nodes in a $k$-hop neighbourhood, where $k$ corresponds to the number of GNN layers.

Oono and Suzuki (2019) investigated the asymptotic behaviour of the expressive power of GCN as the number of layers tends to infinity. They show that the node representations converge to a single signal determined by the node's connected component and degree, as well as the exponential nature of this asymptotic behaviour, which is defined as

$$d_{\mathcal{M}}(\boldsymbol{h}^{(\ell)}) \leq s\,\lambda\,d_{\mathcal{M}}(\boldsymbol{h}^{(\ell-1)}),$$

where $d_{\mathcal{M}}(\boldsymbol{h}^{(\ell)})$ denotes the distance of the node representation $\boldsymbol{h}$ in layers $\ell$ to the mean representation of subspace $\mathcal{M}$ of the same connected component and node degree. This distance measures the smoothness of the graph. The over-smoothing factor $s\lambda$ comprises the non-trivial eigenvalue of the propagation matrix $\lambda$ with the largest absolute value and the largest singular value of all weight matrices $W_\ell$, or in other words, the maximum amplification factor. For most graphs, $s\lambda < 1$, implying inevitable over-smoothing.

Keriven (2022) provided a theoretical analysis of the effect of message passing, focusing not just on the asymptotic behaviour but also the optimal GNN depth. With a simplified theoretical model, he showed that a limited number of message passing layers improves learning due to faster shrinking of non-principal than of principal components and compression of inter-communities variance before complete collapse occurs.

Building on the curvature-based framework for over-squashing introduced by Topping et al. (2022), Nguyen et al. (2023) connected positive edge curvature to over-smoothing. They showed that higher neighbourhood overlap between adjacent nodes implies higher curvature:

$$\frac{|\mathcal{N}(v) \cap \mathcal{N}(u)|}{\max(\deg(v), \deg(u))} \geq \kappa(v, u),$$

and hence contributes to the smoothing of node features.

In a comprehensive survey, Rusch et al. (2023a) compare different metrics and mitigation strategies for over-smoothing empirically. They advocated for the use of *Dirichlet energy* as a measure for smoothness of $\boldsymbol{H}_{\mathcal{G}}^{(\ell)}$ the matrix of all hidden vectors $\boldsymbol{h}_v^{(\ell)}$ in graph $\mathcal{G}$, which has a comparable behaviour to MAD and is defined as

$$\mathrm{E}(\boldsymbol{H}_{\mathcal{G}}^{(\ell)}) = \frac{1}{|\mathcal{V}|} \sum_{i \in \mathcal{V}} \sum_{j \in \mathcal{N}(i)} \left\| \boldsymbol{h}_i^{(\ell)} - \boldsymbol{h}_j^{(\ell)} \right\|_2^2.$$

Further, they categorise existing mitigation approaches into: (1) normalisation and regularisation, (2) change of the GNN dynamics, and (3) architectural enhancement through residual connections. Even though they show empirically the suitability of all three approaches for mitigating over-smoothing, the performance decreases or stagnates on the actual learning task as GNN depth increases.

**Methodology Addressing Over-smoothing.** Several architectural interventions have been proposed to counter over-smoothing. Zhao and Akoglu (2020) introduced PairNorm to reduce the effect of over-smoothing by stabilising the sum of the distances between node pairs with and without edges. Chen et al. (2020) show that simple skip connections can enable deep GCNs. Specifically, they introduced GCNII, which adds the initial embedding to each hidden embedding by a proportion $\alpha$ and an identity mapping to each weight matrix $\boldsymbol{W}^{(\ell)}$ by a proportion $\beta$, where $\alpha$ and $\beta$ are hyperparameters. Scholkemper et al. (2025) proved theoretically that both batch-normalisation and residual connections prevent over-smoothing in GCN. These theoretical findings are supported by empirical results on GNN with the most common message passing methods. Rusch et al. (2023b) further developed the idea of residual layers to gradient gating (G$^2$) by replacing the fixed proportion $\beta$ with learnable element-wise rates $\boldsymbol{\tau}^{(\ell)}$. In the G$^2$ model, $\boldsymbol{\tau}^{(\ell)}$ is given by a learnable function of the hidden features and the graph structure.

The most influential contributions on over-smoothing with experimental results investigate this phenomenon only on node-level tasks, i.e., node classification. The effect on the performance of graph classification has not been explored much to the best of our knowledge.

## 2.2 Position

We question the negative influence of over-smoothing in real-world datasets. We consider two different cases of learning tasks with regard to the possible influence of over-smoothing, node-level tasks and graph-level tasks. For node-level tasks, we suspect the performance decrease on real-world data arises not from over-smoothing, but as a consequence of *uninformative receptive fields*. For most learning tasks, a small receptive field is sufficient to encode the relevant information, which indicates a small problem radius. If the problem radius is smaller or similar to the number of message passing steps to achieve optimal smoothing, as described by Keriven (2022), the performance will not suffer from over-smoothing. We support this position with experimental results in Section 2.3, which show optimal model depths even for methods that diminish the effect of over-smoothing. For graph-level tasks, over-smoothing can even be considered beneficial if the common node representation aligns with the label distribution (Southern et al., 2025).

We furthermore want to specify that the problem of oversmoothing is known to originate from the repeated application of message passing functions. Consequently, one can still fit arbitrarily deep GNNs without encountering oversmoothing, by applying a limited number of message passing functions while working with arbitrarily deep update functions. Since, especially for large graph structures, the implementation of message passing comes at a significant computational expense, we posit that the desire to perform arbitrarily many message passing operations is not aligned with the practical deployment of GNNs.

Finally, we furthermore want to highlight that scaling the performance of GNNs to a very large number of message passing operations, may be misguided due to the methodological alternatives that have been developed over the past years. Specifically, for problems with a very large problem radius, i.e., problems that justify the consideration of a very large receptive field per node, one may almost surely want to explore alternatives such as Graph Transformers (Kreuzer et al., 2021; Ying et al., 2021; Rampášek et al., 2022) or rewiring techniques (Topping et al., 2022; Nguyen et al., 2023) to consider these large receptive fields more efficiently than to perform as many or more message passing operations than the problem radius.

**Outlook.** As a consequence of the questionable practical relevance of over-smoothing, the graph learning community should focus its efforts on problems of proven relevance implicated by the conditions in given learning tasks and datasets. We suggest it might be helpful to develop statistics

Table 1: Comparison of a GCN and three methods to mitigate over-smoothing for a range of model depths. The accuracy is given in %, and the computational time is given in milliseconds in brackets.

| Dataset | Model | Layers | | | | | |
|---|---|---|---|---|---|---|---|
| | | **2** | **4** | **8** | **16** | **32** | **64** |
| Cora | GCN | **85.3** (51) | **85.3** (69) | 80.6 (84) | 60.9 (131) | 30.4 (208) | 15.7 (357) |
| | Pairnorm | 83.4 (54) | **83.7** (69) | 83.1 (91) | 65.1 (137) | 26.0 (229) | 19.2 (386) |
| | GCNII | 86.2 (48) | 87.0 (57) | **87.4** (71) | **87.4** (81) | 87.3 (129) | 87.1 (219) |
| | G$^2$ | **72.3** (74) | 70.8 (96) | 71.4 (135) | 72.1 (229) | 70.6 (396) | 70.8 (682) |
| Citeseer | GCN | **78.3** (50) | 76.3 (60) | 68.7 (75) | 35.7 (76) | 20.7 (149) | 18.4 (291) |
| | Pairnorm | **76.4** (53) | 71.6 (63) | 54.2 (76) | 31.9 (106) | 22.5 (164) | 21.4 (279) |
| | GCNII | 79.1 (47) | **79.3** (58) | 79.0 (75) | 79.1 (104) | 79.2 (169) | 79.2 (221) |
| | G$^2$ | 59.8 (76) | 61.2 (101) | 61.8 (141) | 63.9 (213) | 64.5 (366) | **64.6** (714) |
| Pubmed | GCN | **86.6** (55) | 86.4 (63) | 84.6 (87) | 84.1 (62) | 49.5 (44) | 36.0 (80) |
| | Pairnorm | **88.3**(56) | 87.4 (68) | 84.7 (92) | 84.1 (115) | 82.9 (59) | 72.6 (103) |
| | GCNII | 84.4 (49) | **85.4** (56) | 84.7 (75) | 83.8 (97) | 84.1 (166) | 78.3 (265) |
| | G$^2$ | 74.3 (60) | 77.8 (72) | **79.0** (107) | 70.6 (167) | 73.6 (275) | 78.3 (506) |
| Roman-Empire | GCN | **28.7** (17) | 19.7 (20) | 13.3 (24) | 11.8 (31) | 9.8 (48) | 6.1 (81) |
| | Pairnorm | **42.6** (18) | 22.7 (21) | 16.6 (27) | 12.2 (38) | 9.1 (65) | 8.3 (105) |
| | GCNII | **34.6** (47) | 34.5 (61) | 32.5 (80) | 32.0 (108) | 31.7 (174) | 31.7 (275) |
| | G$^2$ | 50.0 (28) | **50.6** (35) | 50.3 (47) | 49.0 (73) | 48.9 (121) | 58.5 (214) |

that quantify the label relevance of different-sized receptive fields. This quantification of the problem radius in combination with an observation of the optimal smoothing depth should guide our efforts in improving model architectures.

## 2.3 Empirical Findings

We back our position on over-smoothing with experimental results, which show that methods presented in Section 2.1 in most cases prevent the models from over-smoothing, but do not score significantly better results with increasing model depth.

**Set-up.** We tested the performance of four architectures that were introduced to mitigate over-smoothing and compared the results to a regular GCN with a range of model depths on multiple commonly used benchmark datasets. We ran the experiments on the homophilic citation datasets Cora, Citeseer and Pubmed as used in (Yang et al., 2016) and the heterophilic Roman-Empire dataset from (Platonov et al., 2024). Even though these datasets have been criticised for a lack of real-world application (Bechler-Speicher et al., 2025), we apply them here, as they have been used to develop most methods against over-smoothing. On each dataset, we tested a regular GCN (Kipf and Welling, 2017), a GCN with PairNorm (Zhao and Akoglu, 2020), a GCNII (Chen et al., 2020), and a G$^2$-GCN with gradient gating (Rusch et al., 2023b). Each of the models was tested with an exponentially increasing number of message passing layers from 2 to 64 layers. Each model was trained for 500 epochs with a learning rate of 0.001 with 20 random parameter initialisations. We set a fixed hidden dimension of 32 features and used early-stopping for regularisation.

**Discussion.** The accuracy of the tested models, along with the computational time in brackets, is presented in Table 1. In this work, we do not focus on the comparison of the absolute results of the different models. Instead, we want to draw the reader's attention to the influence the model depth has on the model's performance. For the regular GCN, we can observe a drastic performance decrease with increasing number of message passings, as expected due to over-smoothing. The GCNII and the G$^2$ do not show this behaviour. Instead, the model performance is stable or increases slightly with the number of layers. However, in most cases, the optimal model depth remains at eight or fewer layers. Additionally, the performance stabilisation at higher model depth comes with an exponential increase in computational costs.

## 3  Over-squashing

We believe that continuous theoretical work on over-squashing should be guided by measurable problems that limit the performance of GNNs in practical applications. In this section, we base our position on the literature, reviewing different perspectives on the phenomenon. We address these perspectives and, as a consequence, question the relevance of over-squashing for practical applications. We support our position with experimental results showing a low optimal model depth with and without methods for reducing over-squashing.

### 3.1  Related Work

Over-squashing is the well-studied phenomenon of compression of an exponentially growing amount of information into fixed-size node embeddings as the depth of a GNN increases. If a node $v$ is to be affected by features of node $u$ at distance $k$, a GNN requires at least $k$ message passing steps. However, an increasing number of message passings results in an exponentially growing number of messages being sent, leading to a loss of information. In other words, over-squashing can be defined as the inability to losslessly compress a receptive field that grows with the depth of the network in fixed-sized node representations. As a consequence of over-squashing, features from neighbours in different $k$-hops can not be considered jointly at the central node, preventing the modelling of interaction effects. In detail, the phenomenon of over-squashing has been defined in various ways, focusing on different aspects of the problem.

**Theory and Quantification of Over-squashing.**   The over-squashing phenomenon was introduced by Alon and Yahav (2020) with the broadest definition of over-squashing. Based on the exponential growth of the receptive field of a node $v$ with the number of message passing layers

$$|\mathcal{N}_L(v)| = \mathcal{O}(e^L),$$

they conclude an over-squashing of an exponentially growing amount of information while the embedding dimension remains constant. They emphasise the increasing severity of this phenomenon for problems that require long-range information, i.e. with a large problem radius $r$. With the synthetic NeighboursMatch dataset, Alon and Yahav (2020) showed the limitation of GNNs with the most common message passing methods for large problem radii and attributed this to over-squashing. Additionally, they validated their results on a real-world dataset and showed that for a GNN with $L = 8$ and using fully-adjacent layers, which theoretically prevents over-squashing, improved the results practically.

Topping et al. (2022) investigate over-smoothing through *graph curvature*, focusing on bottleneck edges that induce over-squashing locally. They formulate the over-squashing problem between a node $v$ and another node $u$ with $d_G(v, u) = r$ through the derivative of the hidden vector $\boldsymbol{h}_v$, i.e., the *Jacobian*, with respect to a feature $h_u$ as

$$\left| \frac{\partial \boldsymbol{h}_v^{(r+1)}}{h_u^{(0)}} \right| \leq (\alpha \cdot \beta)^{r+1} \left( \tilde{\boldsymbol{A}}^{r+1} \right)_{v,u}, \tag{1}$$

with the layer-wise Lipschitz constants $|\Delta U^{(\ell)}| \leq \alpha$ and $|\Delta M^{(\ell)}| \leq \beta$ for $0 \leq \ell \leq r$. The factor $(\tilde{\boldsymbol{A}}^{r+1})_{v,u}$ causes vanishing gradients, especially with bottleneck edges on the path between the two nodes. Topping et al. (2022) propose a modified Forman curvature, which uses the counts of triangles and 4-cycles based on an edge, to identify these bottlenecks.

Di Giovanni et al. (2023) extended the fundamental understanding of over-squashing by investigating its impact on the width, depth and topology, viewing small Jacobians of node features for long-range interactions in general as a problem of over-squashing. They defined an alternative upper bound for the Jacobian, which differs slightly from Equation (1) and is defined as

$$\left| \frac{\partial \boldsymbol{h}_v^{(\ell)}}{\partial \boldsymbol{h}_u^{(0)}} \right| \leq \underbrace{(c_\sigma \cdot w \cdot p)^\ell}_{\text{model}} \underbrace{\left( \tilde{\boldsymbol{A}}^\ell \right)_{vu}}_{\text{topology}}, \tag{2}$$

where $c_\sigma$ is the Lipschitz constant of the nonlinearity $\sigma$, $w$ is the maximal entry-value over all weight matrices, and $p$ is the hidden dimension or model width. They derive from Equation (2) that a larger hidden dimension prevents over-squashing if it compensates for the topology factor. However, they point out that an increased hidden dimension reduces the ability of generalisation and does not solve the actual problem, the graph topology. Further, they prove for tasks with long-range dependencies, i.e., large problem radii $r$, the occurrence of over-squashing for models with a depth comparable to the problem radius ($L \approx r$) and vanishing gradients for deeper models ($L \gg r$). Lastly, Di Giovanni et al. (2023) add another perspective on over-squashing and show that over-squashing occurs for a pair of nodes $v, u \in \mathcal{V}$ with a high commute time $\pi$, which measures the expected number of steps for a random walk to commute between $v$ and $u$.

**Methodology Addressing Over-squashing.**  Topping et al. (2022) introduce a curvature-based rewiring method incorporating their Balanced Forman metric to reduce over-squashing on bottlenecks. The *Stochastic Discrete Ricci Flow* (SDRF) adds edges to support low-curvature edges and removes high-curvature edges. A similar approach called *Batch Ollivier-Ricci Flow* (BORF) was proposed by Nguyen et al. (2023), using the Ollivier-Ricci curvature, which is defined as

$$\kappa(v, u) = \frac{W_1(\mu_v, \mu_u)}{d_G(v, u)},$$

where $W_1(\mu_v, \mu_u)$ is the L1-Wasserstein distance and $d_G(v, u)$ is the shortest path distance. They use $\kappa$ for adding and removing edges in the graph to simultaneously mitigate over-squashing and over-smoothing, respectively. Deac et al. (2022) create a modified graph structure using a fundamentally sparse family of expander graphs with a low diameter. In the Graph Expander Propagation (EGP), they replace every second propagation step with edges from the expander, reducing effective path lengths. Southern et al. (2025) build upon the observations of Di Giovanni et al. (2023) that over-squashing comes with long-range dependencies with long commute times and show theoretically and empirically that a virtual node can reduce $\pi$ and therefore reduce the effect of over-squashing.

## 3.2 Position

We challenge the presumed detrimental effect of over-squashing in real-world applications. More specifically, we question the practical relevance of joint observations of entire receptive fields, long-range interactions, and information exchange through bottlenecks. Note that while these three definitions of the over-squashing problem may appear disparate, they all define different aspects of a coherent notion of the over-squashing problem. We now discuss our position on the three concrete over-squashing definitions in turn.

The formulation of the over-squashing problem through a limit of the Jacobian is based on the assumption that long-range interactions are generally informative for learning tasks and desirable to achieve through deep GNNs. We believe that in most practical applications, the relevant information on interaction effects is stored within a small $k$-hop neighbourhood.

Next, we challenge the negative effect of low-curvature edges. Bottleneck edges limit the exchange of information between structural communities. An interrelation between negative curvature and the label distribution has not been investigated yet. However, we assume in most cases a correlation between structural communities and the label distribution. Consequently, information exchange along bottleneck edges is not relevant for such learning tasks.

At last, we consider the most general formulation of the over-squashing phenomenon, that is, the exponentially growing receptive field and the presumed impossibility of summarising its information in a constant hidden dimension. Implicit in this definition of oversquashing is the assumption that the information of the entire receptive field needs to be jointly observed to perform the given learning task. In other words, the impossibility definition states that the joint distribution over the receptive field of a node does not factorise into marginal distributions over the nodes, i.e., we assume the presence of high-order interaction effects within the receptive field of a node. We posit that this assumption is unrealistic and that for most real-world datasets, the joint distribution over the receptive field of a node does factorise and that we can hence process subsets of the receptive field independently and efficiently. In essence, it seems to us that a fixed-size node representation should be sufficient to

successfully complete the majority of learning tasks for arbitrarily large receptive fields on real-world datasets.

Our assumptions are supported by experimental results we present in Section 3.3. These indicate that despite effectiveness against over-squashing, the overall performance on the learning task does not improve with the increasing number of message passing steps.

**Outlook.** We suggest a diligent analysis of future learning tasks and datasets for a better understanding of the feature, structure and label distributions and their interplay. Statistics are required to measure the localisation and factorisation of the label distribution conditioned on the structure and feature information. In addition to the relevant problem radius as proposed in Section 2.2, we suggest investigating the *specific localisation* of relevant information, which can help understand interaction along low curvature edges and long-range relationships, to guide targeted rewiring techniques. First attempts have been made to measure this through the calculation of the Jacobian in a preprint (Liang et al., 2025). However, this statistic lacks diagnostic precision due to the overlapping effects of the model definition and underlying structure. To us it seems preferable to separately establish the presence of potential over-squashing effects in the underlying graph structure and in the different models that could be applied to this graph data. We furthermore want to specify that the definition of oversquashing via the Jacobian and the related remedy of rewiring, operate under the assumption that single nodes, often at a large shortest path distance, contain information that is uniquely relevant to accurately represent a considered central node. It seems to us that further research on this problem should formally define structures in which such far-removed nodes do contain information that is relevant to the representation of different nodes in the graph and that real-world examples, in which such phenomena can be observed and measured, should be found. The extension of the excellent work by Liang et al. (2025) to not only record the average Jacobian over $k$-hop neighbourhoods, but to also look for outliers in the Jacobian values in the $k$-hop neighbourhoods could be a starting point for such research, since it may permit the identification of individual nodes at a potentially large distance, which may go unnoticed if one averages over entire $k$-hops (even if as specified earlier, it may be preferable to measure such effects for the used model and underlying graph structure separately).

Moreover, we suggest exploring potential *factorisation* of the label distribution over $k$-hop receptive fields to assess the importance of a joint observation of the whole receptive field. The quantification of interaction effects in receptive fields of nodes has the potential to not only guide the development of future GNNs that are more closely aligned with the challenges posed by impactful, real-world learning problems, but also, to offer an insightful categorisation of existing datasets and the ability of models to capture such effects. The application of these statistics shall allow us to better understand existing work and to set the right focus for further research directions.

### 3.3 Empirical Findings

In this section, we give empirical evidence that shows a significant reduction of over-squashing is possible with the proposed methods, while the overall performance on the learning tasks can benefit from this improvement.

**Set-up.** We explore the potential of over-squashing mitigation techniques on node-level and graph-level tasks. For node classification, we use the same four datasets, Cora, Citeseer, Pubmed, and Roman-Empire, as discussed in Section 2.3. For experiments on graph classification, we use the datasets MUTAG, ENZYMES, and PROTEINS from the TU dataset collection (Morris et al., 2020) with the data splits and test procedure described in (Errica et al., 2020). On each dataset, we compare the performance of a regular GCN with EGP Zhang et al. (2025) and BORF Nguyen et al. (2023). We apply BORF once on the whole dataset before training a regular GCN. To ensure comparability, we also implement EGP with GCN message passing instead of GIN as proposed in the original paper. We use the same training and model setup as described in Section 2.3.

**Discussion.** The results of the three models in terms of accuracy are reported in Tables 2 and 3, along with their respective computational time. As in Section 2.3, we can compare how an increasing number of message passings influences the performance of the models. The results on node classification in Table 2 show contrasting effects of both mitigation techniques. While BORF enables better results with deep GCNs, presumably through the elimination of bottlenecks, EGP optimises the long-range message flow for a minimal number of layers. However, the optimal model depth is 4 or less for all

Table 2: Comparison in node-classification of a GCN and two methods to mitigate over-squashing for a range of model depths. The accuracy is given in %, and the computational time is given in milliseconds in brackets.

| Dataset | Model | Layers | | | | | |
|---|---|---|---|---|---|---|---|
| | | **2** | **4** | **8** | **16** | **32** | **64** |
| Cora | GCN | **85.3** (51) | **85.3** (69) | 80.6 (84) | 60.9 (131) | 30.4 (208) | 15.7 (357) |
| | BORF | **85.0** (13) | **85.0** (15) | 80.5 (19) | 68.7 (29) | 58.6 (44) | 25.6 (72) |
| | EGP | **85.4** (25) | 73.6 (28) | 27.2 (32) | 21.4 (40) | 13.3 (56) | 15.4 (90) |
| Citeseer | GCN | **78.3** (50) | 76.3 (60) | 68.7 (75) | 35.7 (76) | 20.7 (149) | 18.4 (291) |
| | BORF | **78.5** (42) | 76.5 (50) | 73.0 (59) | 67.6 (83) | 62.4 (138) | 21.2 (228) |
| | EGP | **78.4** (30) | 65.6 (32) | 27.8 (36) | 16.3 (44) | 14.3 (59) | 14.1 (95) |
| Pubmed | GCN | **86.6** (55) | 86.4 (63) | 84.6 (87) | 84.1 (62) | 49.5 (44) | 36.0 (80) |
| | BORF | 84.9 (39) | **85.2** (54) | 84.1 (58) | 81.9 (91) | 72.8 (142) | 37.6 (234) |
| | EGP | **86.6** (136) | 86.4 (138) | 71.1 (140) | 42.0 (147) | 36.6 (160) | 37.2 (200) |
| Roman-empire | GCN | **28.7** (17) | 19.7 (20) | 13.3 (24) | 11.8 (31) | 9.8 (48) | 6.1 (81) |
| | BORF | 43.1 (48) | **45.4** (56) | 41.7 (70) | 28.2 (70) | 26.8 (103) | 11.9 (76) |
| | EGP | **28.7** (471) | 12.1 (486) | 8.1 (496) | 7.6 (507) | 7.1 (528) | 7.1 (557) |

Table 3: Comparison in graph-classification of a GCN and two methods to mitigate over-squashing for a range of model depths. The accuracy is given in %, and the computational time is given in milliseconds in brackets.

| Dataset | Model | Layers | | | | | |
|---|---|---|---|---|---|---|---|
| | | **2** | **4** | **8** | **16** | **32** | **64** |
| PROTEINS | GCN | 70.2 (61) | 74.2 (66) | **76.1** (77) | 75.8 (101) | 62.1 (146) | 58.6 (226) |
| | BORF | **73.6** (66) | 65.0 (72) | 60.2 (83) | 60.9 (104) | 59.4 (153) | 60.3 (232) |
| | EGP | **69.6** (267) | 66.7 (273) | 63.6 (283) | 64.5 (307) | 62.3 (352) | 64.6 (433) |
| ENZYMES | GCN | 43.7 (40) | **44.2** (44) | 38.3 (51) | 34.9 (66) | 32.2 (94) | 19.8 (202) |
| | BORF | **46.6** (63) | 43.1 (81) | 35.9 (116) | 33.4 (213) | 23.6 (337) | 21.4 (606) |
| | EGP | 42.2 (144) | **43.5** (147) | 38.3 (155) | 34.9 (170) | 32.2 (202) | 21.7 (254) |
| MUTAG | GCN | 76.4 (25) | 79.1 (29) | 79.8 (35) | **81.5** (50) | 76.8 (78) | 68.8 (132) |
| | BORF | 71.8 (26) | **77.6** (30) | 70.2 (36) | 69.0 (51) | 66.5 (82) | 68.1 (134) |
| | EGP | 73.4 (52) | 76.2 (55) | **76.6** (62) | 72.2 (76) | 70.6 (105) | 71.0 (161) |

models, indicating low relevance of the over-squashing problem. On the graph-classification tasks, model depth appears to be a minor problem in general, as the performance decrease on the GCN is less severe. Here, the GCN can not benefit from rewiring techniques.

# 4 Conclusion

In this position paper, we highlight the importance of an in-depth understanding of real-world learning problems to guide future research directions in theoretical work in graph representation learning. While much excellent work has been done to better understand and solve over-squashing and over-smoothing, the practical relevance of these problems should be questioned. As Bechler-Speicher et al. (2025) very recently criticised the current benchmarking culture and called for new transformative real-world benchmarking datasets, we hope that many new learning tasks with individual challenges will emerge and gain importance. It seems crucial to us that as part of this search for new applications and datasets, we analyse these diligently, especially for topics like over-smoothing and over-squashing. Statistics should be established to measure the localisation and factorisation of the feature, structure and label distributions.

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
