# OpenReview forum: "Position: Graph Learning May Have Been Misled By Over-smoothing And Over-squashing"
_NeurIPS.cc/2025/Position_Paper_Track — Submitted to NeurIPS 2025 Position Paper Track_

### Official Review · Reviewer_CcLF · 2025-08-06

**Significance:** 2
**Presentation:** 3
**Rating:** 5
**Confidence:** 4

**Summary:**

The paper argues a position that graph learning might be misled by oversmoothing and oversquashing. Authors discuss the background works and mitigation techniques on both oversmoothing and oversquashing. They further conducted experiments on the small or medium-scale node-level and graph-level tasks to demonstrate that the effects of oversmoothing and oversquashing may not be the actual reason for performance degradation.

**Strengths:**

1. The presentation of the paper is impressive and very easy to follow.

2. A good amount of background work is discussed for both Oversmoothing and Oversquashing.

3. The experimental analysis is moderate. The corresponding discussions further support the position of the paper.

**Weaknesses:**

1. At lines 153-154, the authors claim that oversmoothing occurs due to the uninformative receptive field, not for stacking multiple message passing layers. This is not true because oversmoothing happens when the receptive field overlaps with increasing the hops. Authors should rephrase or explain in detail their thoughts.

2. Only a few small datasets do not imply that oversmoothing or overquashing has no impact on the graph learning paradigm. Experiments on some OGB datasets can be beneficial.

3. In the experiments, the authors showed that overquashing may not be severe for node and graph-level tasks. I disagree with the empirical results. Refer to the work [1] that extensively analyzes the impact of oversquashing in the molecular graphs, and they resolved it through rewiring.  Furthermore, experiments should also be done on LRGB datasets [2].

I am ready to increase my score if my concerns are addressed.

[1] Barbero, Federico, et al. "Locality-aware graph-rewiring in GNNs." ICLR 2023.

[2] Dwivedi, Vijay Prakash, et al. "Long-range graph benchmark." Advances in Neural Information Processing Systems 35 (2022): 22326-22340.

**Questions:**

Check the weakness section.

**Alternative Position:**

No

**Author Identification:**

No.

**Context:**

3

**Discussion:**

4

**Ethics:**

["NO or VERY MINOR ethics concerns only"]

**Position:**

Yes, the paper argues for or against a position related to machine learning.

**Support:**

2

**Thoroughness:**

3

---

### Official Review · Reviewer_kGga · 2025-08-08

**Significance:** 3
**Presentation:** 3
**Rating:** 5
**Confidence:** 3

**Summary:**

This paper discusses about the well known problems in Graph Neural Networks of over-smoothing and over-squashing. It argues that instead of focusing on the problems, a different approach towards the quantification of the problem is required to develop better suited model architectures.

For over-smoothing, the argument presented in this paper is that the it occurs only with much deeper GNNs which have large receptive fields. Quatifying the receptive field suitable for a problem, is more important than trying to solve over-smoothing. For problems that require smaller receptive fields, over-smoothing will not be a problem since the GNN would not be much deep, and for problems that require larger receptive field, there are alternative architectures like Graph Attention Networks and virtual nodes.

A similar argument has been brought up for over-squashing as well - practically, a small neighbourhood is enough to get all relevant information for the learning task.

**Strengths:**

The paper explores the relevant background and existing methods quite well. It cites various other position papers and works that tried to mitigate the above problems. They have shown empirical results of performance of some baseline models and methods to mitigate over-smoothing and over-squashing with exponentially increasing number of layers. The results support their arguments that designing mitigation methods alone is not enough to tackle these problems, rather a better understanding of the problem is required for better suited model architectures.

**Weaknesses:**

For the empiricla results, they have only compared the mitigation methods with GCN. While the objective was to show the improvement over vanilla grpah convolution, it would be better if some widely used methods were also included, like GraphSAGE and Graph attention networks.

**Questions:**

Check Weakness

**Alternative Position:**

Yes, and alternative positions are trivial straw-man arguments

**Author Identification:**

No.

**Context:**

3

**Discussion:**

3

**Ethics:**

["NO or VERY MINOR ethics concerns only"]

**Position:**

Yes, the paper argues for or against a position related to machine learning.

**Support:**

3

**Thoroughness:**

3

---

### Official Review · Reviewer_1xAR · 2025-08-12

**Significance:** 2
**Presentation:** 3
**Rating:** 3
**Confidence:** 4

**Summary:**

The paper argues that the practical impact of oversmoothing and oversquashing in GNNs is often overstated. For node-level tasks, performance drops are due to uninformative receptive fields rather than oversmoothing. The authors claim that oversquashing is less harmful when relevant information is local and ``factorised'', making deep models and long-range aggregation unnecessary. Empirically, mitigation methods often do not lead to performance improvements when deep models are considered while being computationally costly. The paper calls for refocusing theoretical research on understanding how label-relevant information is distributed in real-world tasks.

**Strengths:**

- The manuscript is well-structured, easy to follow, and accessible to a broad audience in ML.
- The subject is highly pertinent to ongoing research in GNNs. Many papers aim to alleviate oversmoothing and oversquashing.

**Weaknesses:**

Overall, I found the empirical analyses weak:
  - The paper only considers 4 datasets (for node-level tasks), 3 of which are homophilic networks. The authors should expand this selection (e.g., OGB datasets). Regarding oversmoothing, what is the rationale for expecting deeper models to outperform shallower ones on highly homophilic graphs? Also, the authors do not consider established benchmarks such as those in [1], which could provide stronger empirical grounding for claims about oversquashing.
  -  Oversmoothing is assessed only on node-level tasks, leaving out other settings (e.g., graph-level tasks).
  - Prior works have proposed different metrics to assess oversmoothing and oversquashing. None of them are reported in the paper. Thus, the relationship between these metrics and downstream accuracy is not explored.
  - The reported results lack error bars, making it difficult to assess significance.
  - On Empire (the only heterophilic dataset), the highest accuracy is at layer 64, not at layer 4 as denoted in the paper. This conflict withs paper's position.

Finally, the work does not offer theoretical insights or formal analysis, which limits its impact and generalizability.

[1] Long range graph benchmark. NeurIPS 2022.

**Questions:**

See weaknesses.

**Alternative Position:**

Yes, and alternative positions are well-considered and named but not addressed

**Author Identification:**

No.

**Context:**

2

**Discussion:**

2

**Ethics:**

["NO or VERY MINOR ethics concerns only"]

**Position:**

Yes, the paper argues for or against a position related to machine learning.

**Support:**

1

**Thoroughness:**

4

---

### Note · Authors · 2025-09-04

**1-10 Additional Comments:**

We have no additional comments.

**1-11 Submit Again:**

Probably yes

**1-1 Submission Process:**

5

**1-2 Next Year:**

A clearer outline of the review and discussion process on the “Call for Position Paper” site.

**1-3 Future Development:**

Having a rebuttal process would be helpful in some cases, where experiments are used to back the position.

**1-4 Interest:**

["Panel discussions with other position paper authors", "Structured debates on controversial topics", "Workshops for developing position papers"]

**1-4 Other Interest:**

No other interests.

**1-5 Thoughtful:**

8

**1-6 Supportive:**

6

**1-7 Technical Aspects Versus Position:**

3

**1-8 Gate Keeping:**

10

**1-9 Camera Ready Changes:**

The reviewers have broadly lauded the clarity of presentation, the breadth of the background material discussed, and the pertinence of our chosen subject. The great majority of the highlighted weaknesses request further experiments, which we have largely conducted. The results, which further strengthen our position, are reported in Section 3 of this survey and will be included in the camera-ready version. The empirical results will cover:

- Results for the OGBN-Arxiv dataset;
- Results for the Peptides-func dataset;
- Results for all our considered datasets and methods with GATv2 and GraphSAGE backbones, excluding GCNII;
- Standard deviations for all results over 10 random parameter initialisations.

Furthermore, we will discuss the most recent publications, which have been published after the submission of our manuscript, to ensure that the review of theoretical works remains up to date. This will include:

- Reference to the metrics proposed by [1], measuring the quality of benchmark datasets. In our position, we call for metrics that complement these to measure the practical relevance of certain theoretical problems of GNNs.
- An additional theoretical perspective on over-smoothing from an optimisation point of view [2].
- An additional recent publication on using the Jacobian for measuring over-squashing [3].
- A position paper that addresses some inconsistencies of the concept of over-smoothing and over-squashing [4]. In [4], the authors take a more conceptual, more theoretical approach than we do in our position, which is more strongly grounded in phenomena present in real-world graphs.

[1] Coupette et al., “No metric to rule them all […]” ICML July 2025
[2] Keriven, “Backward oversmoothing: why is it hard to train deep graph neural networks?” arXiv May 2025
[3] Bamberger et al., “On Measuring Long-Range Interactions […]” ICML July 2025
[4]  Arnaiz-Rodriguez and Errica, “Oversmoothing, ‘Oversquashing’, Heterophily, Long-Range, and more […]” arXiv June 2025

**3-1 Review Response1:**

1xAR

**3-2 Reaction To Review1:**

W1: We conducted experiments on an OGB and an LRGB dataset, as suggested by the reviewer. The results, reported in the tables below, further strengthen our position.

Accuracy for OGBN Arxiv (169k nodes, 1.1M edges).

|Model|2|4|8|16|32|64|
|---|---|---|---|---|---|---|
|GCN|**50.0**|43.9|25.9|16.9|13.5|13.1|
|Pairnorm|**57.9**|57.1|46.1|56.3|41.2|19.1|
|GCNII|53.6|**55.6**|55.4|55.4|55.3|54.7|

Average Precision for Peptides-func (15.5k graphs, 2.3M nodes).

|Model|2|4|8|16|32|64|
|---|---|---|---|---|---|---|
|GIN|59.3|**60.7**|57.9|60.7|55.7|42.2|
|BORF|59.0|**59.7**|57.9|55.8|49.7|47.0|
|EGP|41.6|**42.9**|21.4|17.5|17.3|17.2|

To also address the reviewer’s question, the expectation of deeper GNNs outperforming shallower ones, originates from deep learning research in other data domains, where deeper models typically achieve better performance. We question the prominent assumption that over-smoothing and over-squashing are responsible for the poor performance of deep GNNs. We do not take a position on heterophily in this paper and would prefer to keep this largely out of scope.

W2: Studying over-smoothing in graph-level tasks is uncommon and would not be reflective of the approaches concerned by our position. In some instances, over-smoothing can even be beneficial in graph-level tasks [1].

W3: We discuss metrics for over-smoothing and over-squashing in Sections 2.1 and 3.1 of our paper. The concurrent position paper [2] explores several problems with these metrics.

W4: We repeated all experiments with 10 random parameter initialisations to record the standard deviations as requested by the reviewer. The results do not affect our position. We will include these in the camera-ready version.

W5: After rerunning the experiments to obtain standard deviations, this specific case turned out to be a statistical anomaly.

[1] Southern et al., “Understanding Virtual Nodes […]” ICRL 2025
[2] Arnaiz-Rodriguez and Errica, “Oversmoothing, ‘Oversquashing’ […]” arXiv 2025

**3-3 Review Response2:**

kGga

**3-4 Reaction To Review2:**

We thank the reviewer for the accurate and insightful summary of our position.

As the reviewer suggested, we have run experiments on most mitigation techniques in combination with GATv2 and GraphSAGE across three datasets. The results reinforce our position regarding the limited relevance of over-smoothing and over-squashing, as the overall best performance per dataset can be achieved with a small number of message-passings (≤8).

|Dataset|Model|2|4|8|16|32|64|
|---|---|---|---|---|---|---|---|
|Cora|GATv2|**74.6**|74.0|39.2|31.9|31.9|31.9|
|Cora|GraphSAGE|**78.9**|73.4|31.9|31.9|31.9|31.9|
|Cora|Pairnorm (GATv2)|**70.6**|61.3|63.1|66.1|49.2|38.7|
|Cora|Pairnorm (GraphSAGE)|**69.7**|47.8|42.2|42.1|37.9|35.6|
|Cora|G2 (GATv2)|76.0|76.9|76.6|75.8|72.6|**78.8**|
|Cora|G2 (SAGE)|75.5|75.2|75.4|74.9|76.3|**78.8**|
|Cora|EGP (GATv2)|**80.1**|31.9|31.9|31.9|31.9|31.9|
|Cora|EGP (GraphSAGE)|**78.3**|31.9|31.9|31.9|31.9|31.9|
|Cora|GATv2 BORF|**78.1**|74.9|40.0|31.9|31.9|31.9|
|Cora|GraphSAGE BORF|**76.7**|73.4|36.0|31.9|31.9|31.9|
|Roman-Empire|GATv2|**33.2**|29.5|14.0|13.9|13.9|13.9|
|Roman-Empire|GraphSAGE|**54.9**|53.8|36.4|13.9|13.9|13.9|
|Roman-Empire|Pairnorm (GATv2)|**43.9**|32.3|21.7|17.2|14.3|13.8|
|Roman-Empire|Pairnorm (GraphSAGE)|57.4|**58.3**|57.1|55.4|36.2|14.1|
|Roman-Empire|G2 (GATv2)|55.2|56.5|55.9|**56.9**|50.4|43.6|
|Roman-Empire|G2 (SAGE)|57.8|56.6|**58.0**|52.9|50.2|54.6|
|Roman-Empire|EGP (GATv2)|**52.0**|24.1|13.9|13.9|13.9|13.9|
|Roman-Empire|EGP (GraphSAGE)|**55.5**|50.9|26.1|13.9|13.9|13.9|
|Roman-Empire|GATv2 BORF|**37.0**|30.3|14.6|14.5|13.9|13.9|
|Roman-Empire|GraphSAGE BORF|**55.5**|55.5|34.5|13.9|13.9|13.9|
|Proteins|GATv2|66.9|67.7|**69.9**|66.8|59.7|57.2|
|Proteins|GraphSAGE|68.1|69.4|68.5|**70.1**|61.8|59.8|
|Proteins|EGP (GATv2)|65.4|**70.1**|66.4|59.6|58.5|59.8|
|Proteins|EGP (GraphSAGE)|**67.6**|63.2|55.2|54.8|55.8|43.7|
|Proteins|GATv2 BORF|67.2|67.3|**69.4**|62.1|52.3|48.0|
|Proteins|GraphSAGE BORF|67.9|68.5|**70.3**|68.5|61.1|59.8|

**3-5 Review Response3:**

CcLF

**3-6 Reaction To Review3:**

We thank the reviewer for the constructive feedback and the recognition of the strengths of the manuscript despite their disagreement with our position. We are delighted to see the reviewer indicating their readiness to increase their score and believe that our answers below merit such an increase.

W1: We believe the reviewer may have misunderstood our statement. In Lines 153–154, we do not claim that over-smoothing arises as a consequence of uninformative receptive fields. We state that the observed performance decrease originates from uninformative receptive fields rather than from the over-smoothing problem. This should resolve the reviewer's concern.

W2: We are confident that the results on OGBN-Arxiv and Pepdtides-func address the reviewer’s concern about the validity of our statements due to the small size of the datasets. These are two of the most widely used graph datasets for node-level and graph-level predictions at scale. The results reported in Tables 1 and 2 further support our position.

Table 1: Accuracy for OGBN Arxiv (169k nodes, 1.1M edges).

|Model|2|4|8|16|32|64|
|---|---|---|---|---|---|---|
|GCN|**50.0**|43.9|25.9|16.9|13.5|13.1|
|Pairnorm|**57.9**|57.1|46.1|56.3|41.2|19.1|
|GCNII|53.6|**55.6**|55.4|55.4|55.3|54.7|

Table 2: Average Precision for Peptides-func (15.5k graphs, 2.3M nodes).

|Model|2|4|8|16|32|64|
|---|---|---|---|---|---|---|
|GIN|59.3|**60.7**|57.9|60.7|55.7|42.2|
|BORF|59.0|**59.7**|57.9|55.8|49.7|47.0|
|EGP|41.6|**42.9**|21.4|17.5|17.3|17.2|

W3: We followed the reviewer’s suggestion and conducted experiments on the Peptides-func dataset, which is part of LRGB. The results are reported above in Table 2. Further, we recognise that “Locality-aware Graph Rewiring” is an additional mitigation technique against over-squashing. However, given the position paper track’s focus on argumentative discussion, we believe that our paper stands on sufficient empirical evidence, including other state-of-the-art rewiring techniques.

---

### Meta-Review · Area_Chair_bC1L · 2025-09-12

**Rating:** 5
**Confidence:** 3

**Strengths:**

The reviewers stated that this is a well-written paper, on a relevant topic, with a well-explained background. The paper has a clear position revealing a disconnect between theory and practice, which is supported by a limited set of experiments.

**Weaknesses:**

The main criticism is that the experiments are limited. They claim that the experiments are not sufficient to justify the paper's position.

**Questions:**

Can you define what you mean by the factorisation of the label distribution?

**Thoroughness:**

1

---

### Decision · Program_Chairs · 2025-09-26

Reject